# Imidazopyridine Family: Versatile and Promising Heterocyclic Skeletons for Different Applications

**DOI:** 10.3390/molecules29112668

**Published:** 2024-06-05

**Authors:** Giorgio Volpi, Enzo Laurenti, Roberto Rabezzana

**Affiliations:** Department of Chemistry, University of Turin, Via Pietro Giuria 7, 10125 Torino, Italy; enzo.laurenti@unito.it (E.L.);

**Keywords:** imidazopyridine, imidazoisoquinoline, imidazo[1,5-*a*]pyridines, imidazo[1,2-*a*]pyridines, imidazo[4,5-*c*]pyridines, imidazo[5,1-*a*]quinolones, imidazo[2,1-*a*]isoquinolines heterocycle

## Abstract

In recent years, there has been increasing attention focused on various products belonging to the imidazopyridine family; this class of heterocyclic compounds shows unique chemical structure, versatile optical properties, and diverse biological attributes. The broad family of imidazopyridines encompasses different heterocycles, each with its own specific properties and distinct characteristics, making all of them promising for various application fields. In general, this useful category of aromatic heterocycles holds significant promise across various research domains, spanning from material science to pharmaceuticals. The various cores belonging to the imidazopyridine family exhibit unique properties, such as serving as emitters in imaging, ligands for transition metals, showing reversible electrochemical properties, and demonstrating biological activity. Recently, numerous noteworthy advancements have emerged in different technological fields, including optoelectronic devices, sensors, energy conversion, medical applications, and shining emitters for imaging and microscopy. This review intends to provide a state-of-the-art overview of this framework from 1955 to the present day, unveiling different aspects of various applications. This extensive literature survey may guide chemists and researchers in the quest for novel imidazopyridine compounds with enhanced properties and efficiency in different uses.

## 1. Introduction

The imidazopyridines are heterocyclic compounds formed by the fusion of an imidazole ring with a pyridine ring. Figure 1 illustrates the various possible combinations of the imidazole and pyridine rings, resulting in different nuclei, all identified and documented in the literature as imidazopyridines. These compounds are primarily distinguished from each other by their geometric arrangements and the bonding between the two aromatic heterocycles.

According to current nomenclature, imidazopyridines are generally categorized into various groups based on the position of the nitrogen atoms and the distinct connections between the imidazole and pyridine rings (refer to Figure 1). Despite potential variations in composition or nitrogen atom count across individual structures (e.g., imidazo[1,2-*a*]pyridine and imidazo[4,5-*c*]pyridine), they all fall under the classification of imidazopyridines [1,2,3,4]. Each imidazopyridine nucleus differs in its peculiar biological, physical, and chemical properties, and these diverse behaviors imply various fields of application [5,6,7].

Furthermore, imidazopyridines can be modified by the insertion of functional groups to alter their properties; the accessibility of each structural position varies depending on the specific family being considered.

Among the different families, the recent scientific literature highlights three main application areas for these promising aromatic heterocycles:Imidazo[1,2-*a*]pyridines are principally studied and applied in pharmacology and imaging, as modulators of the central nervous system, and as a biologically active core for the development of various commercial drugs [8,9,10,11,12].Imidazo[1,5-*a*]pyridines are investigated and utilized for their emissive properties (serving as fluorophores) [13,14,15] or chelating ligands for metallic ions, contributing to the development of on–off-type sensors [16,17,18,19,20,21,22].Imidazo[4,5-*b*]pyridines are principally studied for the development of emissive products for OLEDs (organic light-emitting diodes) and sensor applications due to their excellent emissive properties [23,24,25,26,27,28].Imidazo[4,5-*c*]pyridines display a bioisosteric resemblance to the purine nucleus, manifesting similar structural and electronic properties. This structural affinity facilitates imidazopyridine’s facile interaction with macromolecules such as DNA, RNA, and specific proteins. Moreover, imidazo[4,5-*c*]pyridines exhibit notable cytotoxic activity by selectively inhibiting kinase activity [29,30,31].

Moreover, imidazoquinolines correspond to various structures (such as imidazo[5,1-*a*]quinolines and imidazo[2,1-*a*]isoquinolines) in which a third aromatic ring connects to the imidazopyridine nucleus. These derivatives represent the extension of the aromatic system of the classic imidazopyridines; the extended delocalized system alters their optical, structural, and solubility properties, sometimes presenting a promising structural alternative to the common heterocyclic ring such as phenanthroline, carbazole, benzoquinoline, acridine, and many others [32,33,34,35,36,37,38,39,40,41,42].

Numerous recent publications have underscored new synthetic strategies leading to promising optical properties, compelling many researchers to delve deeper into exploring this class of emissive compounds. Thanks to innovative methodologies, they have introduced various chemical groups aimed at conjugation and modulation of optical properties [5,43,44,45,46].

In recent years, many works focused on imidazopyridine derivatives have been published, describing different synthetic cyclization strategies or structural modifications. Generally, the imidazopyridine skeleton has been designed to obtain structural derivatives such as luminescent probes and ligands for transition metal ions or to increase pharmaceutical and biological activities. In particular, some works were focused on the connection of imidazopyridine with other fluorophores (such as boron-dipyrromethene, BODIPY or boron-dipyrromethene, and rhodamine) or on their activation as the fluorogenic center itself [8,47,48,49].

In general, the distinctive emission of the imidazopyridine derivatives is commonly found in the range of 430–520 nm, with a broad Stokes shift (up to 80 nm) and high values of quantum yield (ranging from 5% to 60%). Due to their structural flexibility and tunability of optical properties, imidazopyridine fluorophores are excellent candidates for various technological applications such as downshifting and optoelectronic devices [50,51,52]. G. Albrecht et al. recently investigated the electroluminescence of imidazopyridine derivatives utilizing layered OLEDs constructed with these emitting molecules [39,40,53,54]. Until now, only copper complexes derived from imidazopyridine ligands have proven effective in light-emitting electrochemical cells [55,56,57].

Moreover, imidazopyridine-based emitters offer a compelling alternative for attaining robust emission in fluorescence-based microscopy and confocal microscopy. Typically, imidazopyridine derivatives demonstrate favorable solubility, intense emission, high quantum yield, pH sensitivity, and strong biocompatibility. In recent years, imidazopyridine fluorophores have been successfully tested for obtaining detailed images through fluorescence microscopy and confocal microscopy [5,58,59,60,61,62,63,64].

Moreover, imidazopyridines present other areas of interest beyond their use as fluorophores. Indeed, among aromatic heterocycles, imidazopyridines have experienced a significant surge in interest in the literature due to their wide variety of applications in pharmaceutical chemistry. Although structurally different, imidazopyridines exhibit pharmacological properties quite similar to benzodiazepines [37,65,66,67]. Compounds containing the imidazopyridine moiety exhibit a diverse array of captivating biological characteristics, including antifungal, antipyretic, antiviral, analgesic, antibacterial, anticancer, antiprotozoal, anti-inflammatory, anticonvulsant, anthelmintic, antitubercular, antiulcer, antiepileptic, hypnotic, and anxiolytic properties [18,68,69,70,71,72]. They have also been the subject of investigation as receptor agonists. The imidazopyridine nucleus, in particular the imidazo[1,2-*a*]pyridines, is also present in several commercially available drugs, as reported in Figure 2, such as zolpidem, alpidem, saripidem, necopidem, olprinone, miroprofene, zolimidine, and many others [28,66,73,74,75,76,77,78,79,80,81].

Therefore, it is not surprising that the number of scientific articles related to the synthesis and application of this family of aromatic heterocycles has exponentially increased in the last decade. In recent years, notable advancements have been made in the heterocyclization and structural alteration of imidazopyridine derivatives through diverse methodologies, including, among others, multicomponent reactions, tandem sequences, transition metal-catalyzed C-H functionalizations, and one-pot syntheses [5,7,12,46,82,83]. These methods offer easy access to imidazopyridine products and their functionalizations starting from simple and readily available precursors.

The objective of this article is to compile and present the documented applications of numerous compounds belonging to the imidazopyridine family. Instead of focusing solely on a specific heterocyclic nucleus, the aim is to offer a comprehensive overview of this promising heterocyclic family, which holds potential for diverse biological and technological applications [6,45,84,85]. The target audience includes researchers already engaged in this field, as well as newcomers seeking insights.

## 2. Studies and Applications of Imidazopyridine Derivatives

### 2.1. Imidazopyridines in Medicine

A detailed survey of the literature has revealed that various classes of imidazopyridines can act as potent modulators toward several pathologies associated with central nervous system dysfunction, including Parkinson’s and Alzheimer’s diseases, depression, schizophrenia, or sleep disorders. The recent literature described in detail the interaction of imidazopyridine derivatives with different enzymes (e.g., leucine-rich repeat kinase, β-secretase, fatty acid amide hydrolase, γ-secretase), receptors (e.g., adenosine A2A, GABA-A (γ-Aminobutyric acid), histamine, serotonin H3, 5-HT35-HT4, 5-HT6, and dopamine D4), and their therapeutic role [16]. In the field of central nervous system (CNS) modulators, the era of imidazopyridines began in the late 1980s and further developed starting in 1992 with the synthesis and approval of the drug zolpidem based on the imidazopyridine nucleus [66,86]. In the subsequent decades, numerous other imidazopyridine-based drugs have been introduced to the market (see Figure 2). Recent research has concentrated on molecular targets within the central nervous system (CNS), specifically linked to diverse neurodegenerative conditions and psychiatric disorders, where imidazopyridines demonstrate intriguing modulation potential [16,65,87,88,89].

Certainly, up to now, the most investigated and utilized imidazopyridines are associated with the imidazo[1,2-*a*]pyridine nucleus, particularly for pharmaceutical and medical applications [84,85,90]. These derivatives exhibit a wide range of biological activities, such as cardiotonic, antifungal, anti-inflammatory, antipyretic, analgesic, antitumor, antiapoptotic, hypnotic, antiviral, antibacterial, antiprotozoal, and anxiolytic agents. Moreover, they act as inhibitors of β-amyloid formation, agonists of GABA, and benzodiazepine receptors [1,37,66,91,92,93]. Several drugs based on imidazo[1,2-*a*]pyridine derivatives are available in the market, such as zolpidem (used in the treatment of insomnia), alpidem (an anxiolytic agent), olprinone (for the treatment of acute heart failure), zolimidine (used for the treatment of peptic ulcer), necopidem, and saripidem (both acting as anxiolytic agents), all containing the imidazo[1,2-*a*]pyridine unit. Additionally, GSK812397 is a drug for HIV infection treatment, and the antibiotic rifaximin also contains this peculiar heterocyclic skeleton [7,43,45,94].

Different imidazo[1,2-*a*]pyridine derivatives exhibit excellent brain permeability and ideal metabolic stability. These data strongly suggest that imidazo[1,2-*a*]pyridine derivatives labeled with iodine-123 can be efficiently employed as potential imaging agents for single-photon emission computed tomography (SPECT) to detect amyloid pathology in the brains of patients with Alzheimer’s disease [17,95]. Recent studies have demonstrated that these compounds exhibit peculiar anticancer activity and could be valuable in developing more effective drugs with antiproliferative activity against lung, breast, and cervical cancer cells [96,97,98,99,100].

Among the pharmacologically relevant imidazopyridines, the group of imidazo[1,5-*a*]pyridines, despite being mostly employed for their luminescent features, has yielded interesting results. Currently, there are no commercial drugs containing this unit, but many studies have demonstrated the potential applicability of imidazo[1,5-*a*]pyridines in pharmacology. In Figure 3, diverse biologically active derivatives of imidazo[1,5-*a*]pyridine are depicted, including agonists of cannabinoid receptor type 2 (CB2R) (compound **1**), serotonin 5-hydroxytryptamine (5-HT4) antagonists (compound **2**), inhibitors of hypoxia-inducible factor 1α (HIF-1α) (compound **3**), inhibitors of indoleamine 2,3-dioxygenase (IDO) and tryptophan 2,3-dioxygenase (TDO) (compound **4**), cribrostatin-6 (compound **5**), phosphodiesterase 10A inhibitors (compound **6**), tubulin polymerization inhibitors (compound **7**), neurokinin antagonists, kinase inhibitors, and various chemotherapeutic agents that have been evaluated [65,75,76,87,92,101,102,103,104,105,106,107]. In addition, in the past, this nucleus has been investigated as a selective, nonsteroidal inhibitor of aromatase for the treatment of estrogen-dependent disease [108].

Other important examples of imidazopyridines with significant pharmacological implications are as follows:Imidazo[1,2-*a*]quinoline was investigated for its antitumor activity, DNA binding, and antitumor evaluation [109,110,111].Imidazo[4,5-*b*]pyridine described in a review focused on the synthesis and pharmacological use of this particular imidazopyridine [30], in particular as an inhibitor of the B-Raf kinase [112].Imidazo[4,5-*c*]pyridines were investigated for their antitumor activity [31] and notable cytotoxic, antiviral, and antimicrobial activity, which are inhibitors of Bruton’s tyrosine kinase enzyme activity [113,114,115].Imidazo[1,2-*b*]pyridazines represent an important class of heterocyclic nuclei that provide various bioactive molecules. Among these, the successful kinase inhibitor ponatinib has sparked renewed interest in exploring new imidazo[1,2-*b*]pyridazine derivatives for their potential therapeutic applications in medicine, including anticancer agents, diagnostic tools for neuropathic diseases, thymic enhancers, antiparasitic, antibacterial, and antiviral agents, anti-inflammatory agents, and treatments for circadian rhythm sleep disorders [116,117,118,119].

### 2.2. Imidazopyridines as Ligands toward Different Metal Ions

Imidazopyridines are versatile and unique molecules in terms of their coordination toward metal ions. They are easily functionalizable by introducing substituents in different positions and can act as ligands toward different metal ions showing varying geometries and bonding angles. The optoelectronic properties of these imidazopyridine derivatives are noteworthy, as compounds derived from this nucleus display distinctive photophysical traits and robust stability under various experimental conditions. Consequently, this class of heterocyclic derivatives is garnering interest from researchers across diverse fields of study worldwide.

While imidazo[1,2-*a*]pyridines have found significant applications as drugs or in the treatment of various pathologies, imidazo[1,5-*a*]pyridines have attracted more interest concerning the formation of organometallic complexes and the study of the optical properties of these products and their applications. In general, extensive literature on transition metal complexes with imidazole or pyridine nuclei is available. Similarly, the imidazopyridinic skeleton can coordinate a wide variety of transition metal ions, yielding numerous metal complexes with diverse coordination motifs.

The metal–ligand chemistry associated with these heterocycles has been extensively studied, involving the formation of metal complexes featuring bidentate or tridentate structures. Various imidazopyridines have been effectively incorporated at numerous positions along a central aromatic ring, a critical structural characteristic ensuring geometrical equivalency with the more extensively researched multi-pyridinic or multi-bipyridinic ligands in coordination chemistry [120]. Multi-imidazopyridines have already been employed as ditopic ligands in binuclear complexes [48,121,122,123,124,125,126,127,128].

The imidazo[1,5-*a*]pyridinic skeleton, serving as a ligand, introduces a novel polyazine framework featuring diverse and adjustable coordination motifs adaptable for mono-, di-, and tritopic coordination sites. These molecules exhibit potential as polydentate ligands or precursors for supramolecular designs. Furthermore, the phenylimidazole unit has been observed to function as a cyclometalating ligand, akin to the conventional 2-phenylpyridine. In this scenario, a pendant phenyl group is appended at position 1 on the imidazolic nucleus, yielding a suitable C-N ligand, successfully employed with various metals (see Figure 4 for complete numeration of the imidazopyridine nuclei).

To date, there are no reported instances of utilizing the imidazo[1,5-*a*]pyridinic skeleton as a C-N ligand, despite its structural resemblance to phenylimidazole or the more traditional 2-phenylpyridine. Typically, the nitrogen of the imidazole ring can act as a monodentate ligand, like common pyridine, albeit with only limited binding capacity and stability. It is indeed the substituted imidazopyridines at positions 1 or 3 with suitable substituents that represent the best ligands obtained from this class of compounds.

Similarly, the imidazo[1,5-*a*]pyridinic nucleus, substituted at position 1 with a pendant pyridinic group, has been utilized for coordinating transition metal ions as a chelating N-N ligand (see Figure 5) [125,129,130,131]. Specifically, the inclusion of a pendant pyridinic group at position 1 on the imidazo[1,5-*a*]pyridinic skeleton (as opposed to introducing, for instance, a pendant phenyl group) enhances the quantum yield values [132]. Moreover, the notable presence of the pendant pyridinic group ensures the characteristic bidentate N-N ligand ability [126,133,134,135,136,137]. This motif is widely recognized for facilitating stable complexation with diverse metals.

In a different scenario, the imidazo[1,5-*a*]pyridinic nucleus can be employed as an N-O ligand if an appropriate substituent is previously introduced at position 1 on the imidazo[1,5-*a*]pyridinic unit (see Figure 5); this type of ligand has been coordinated to Ni, Co, and Pd, with the purpose to prepare emissive products for optoelectronic devices [134,135,138].

Furthermore, significant effort has been directed towards the advancement of luminescent boron complexes, such as BODIPY, the most renowned luminescent organo-boron compound. However, only a limited number of boron complexes based on the N-O chelating imidazopyridine ligand have been documented [8,20,122,137,139].

Zinc complexes have been successfully synthesized to produce fluorescent materials for optoelectronic devices. In each instance, a comparison of the optical properties between free imidazopyridinic ligands and their corresponding Zn(II) complexes reveals a notable enhancement in quantum yield and an intriguing hypsochromic shift. This shift arises from structural alterations in the conformation of imidazo[1,5-*a*]pyridinic ligands during zinc coordination and subsequent complexation reactions [36,125,136,138,140,141,142,143].

In a prior study focusing on the catalytic dimerization of ethylene, nickel complexes based on imidazo[1,5-*a*]pyridine derivatives were synthesized and characterized [144,145,146].

Copper complexes have undergone extensive investigation in OLED and LEC (Light-emitting Electrochemical Cell) devices, showcasing notable yellow electrochemical emission in cells employing heteroleptic Cu(I) complexes emitting in blue [55]. Despite this, the development of new Cu(I) complexes for blue LEC design remains largely unexplored. In this context, imidazopyridinic derivatives emerge as a promising alternative to complexes based on the more widely recognized bipyridine and phenanthroline [39,40,53,54,55,57,147,148].

Iron and cobalt complexes based on imidazopyridine have been synthesized and structurally characterized [149,150,151,152,153]; initial findings suggest the spontaneous resolution of enantiomers during crystallization, indicative of a rare p-donor/p-acceptor chiral recognition mechanism [154].

The design and self-assembly of chiral complexes have garnered significant interest due to their relevance in various enantioselective processes such as asymmetric catalysis, chemical sensing, and selective guest inclusion. Although spontaneous resolution is considered a more economical and convenient method, it is relatively rare and less predictable because the underlying conglomeration processes are not fully understood.

Imidazo[1,5-*a*]pyridinic chelating ligands have been effectively utilized to form complexes with iridium and rhenium, resulting in emissive products [155,156,157,158]. Computational calculations in these instances reveal emission primarily centered on the ligand, displaying minimal charge transfer character or dual emission. The employment of DFT (Density Functional Theory) and TDDFT (Time-dependent Density Functional Theory) calculations has played a pivotal role in elucidating the distinctions among these imidazo[1,5-*a*]pyridine-based complexes, their structures, and excited states. These investigations underscore the potential of Re and Ir complexes for optical applications, as the excited states of their metal complexes can be readily manipulated by altering substituents on the imidazo[1,5-*a*]pyridinic ligand nucleus [10,155,157,159]. Finally, cadmium, osmium, and ruthenium have been successfully complexed with imidazo[1,5-*a*]pyridinic-based ligands, yielding interesting products suitable for supramolecular chemistry [133,160,161,162,163].

Currently, no examples of complexation of other imidazopyridine derivatives have been reported, except for imidazo[1,2-*a*]pyridines, for which some references are available concerning ruthenium complexes used as catalysts [164] and highly emissive zinc complexes [165,166,167].

### 2.3. Imidazopyridines Applications in On–Off-Type Sensors

Detecting and monitoring hazardous contaminants like pesticides, metals, and anions within acceptable environmental and cellular limits is crucial for assessing environmental quality (air, water, soil) and understanding their harmful effects on living organisms. The health of organisms is directly or indirectly impacted by these contaminants. Consequently, developing sensors capable of detecting such contaminants is vital for protecting ecosystems and living organisms. To this purpose, among the various sensing techniques employed, optical methods based on fluorescence properties represent an excellent approach. Fluorescent probes have gained attention due to their ability to monitor molecular interactions in real time with high sensitivity, selectivity, direct visualization, and short response time, and without interfering with biological samples. A wide array of fluorescent probes, including naphthalimide, fluorescein, BODIPY, coumarin, rhodamine, cyanine, and, more recently, imidazopyridine-related examples, are available for this purpose [8,48,168,169].

Thennarasu and his group reported a chemosensor based on the imidazo[1,2-*a*]pyridinic nucleus for the sequential detection of Cu^2+^ and CN^−^ ions using fluorimetry [26]. The studied compound exhibits a characteristic absorption band at 322 nm. Upon the addition of various metal ions, a slight shift in the fluorescence spectrum was observed, whereas the Cu^2+^ ion nearly completely quenched the fluorescence. Subsequently, among the various anions tested, only CN^–^ caused the turn-on fluorescence of the imidazo[1,2-*a*]pyridinic-Cu^2+^ complex produced. Competition experiments between metal cations and anions showed a complete lack of potential interference in the detection and quantification of Cu^2+^ and CN^–^ ions. It has also been demonstrated through in vivo experiments that this chemosensor is suitable for fluorescence detection of Cu^2+^ and CN^–^ ions in human urine and blood [32,170,171]. Following this initial example, several others on similar compounds have been reported and are documented in the review dedicated to chemosensors based on imidazo[1,2-*a*]pyridines [5].

Similarly, derivatives of imidazo[1,5-*a*]pyridines, known for their intense luminescence, have been widely reported as a basis for new chemosensors. Hou et al. reported a robust ratiometric fluorescent probe for glutathione detection based on imidazo[1,5-*a*]pyridine fluorophore [172]. R. Mayilmurugan et al. reported an interesting study regarding the application of imidazo[1,5-*a*]pyridine derivatives as sensors for cysteine [173]. Generally, cysteine overexpression is well known in a wide variety of different kinds of tumors. In their research, the authors developed and prepared novel Cu(II) complexes utilizing imidazo[1,5-*a*]pyridine derivatives as optical chemosensors, employing a “turn-on” luminescent mechanism for cysteine detection. The investigated probe demonstrated distinct and intense emission upon interaction with cysteine, enabling imaging of tumor cells through the in situ reduction of copper (from Cu(II) to Cu(I)).

Another intriguing application of chemosensors derived from imidazo[1,5-*a*]pyridines involves the detection of hydrogen polysulfides (H_2_S*_n_*), offering the potential for biomedical applications [174]. In general, hydrogen polysulfides play a pivotal role in various critical biological functions and processes associated with hydrogen sulfide. Despite their significance, developing probes capable of rapidly, selectively, and sensitively detecting hydrogen polysulfides poses a considerable challenge. S. Wang et al. have addressed this challenge by designing, synthesizing, and effectively applying a novel probe based on the imidazo[1,5-*a*]pyridinic structure. This probe offers a wide Stokes shift, a low detection limit, and minimal cytotoxicity. Furthermore, this newly developed lysosome-targeted probe has proven successful in detecting endogenous hydrogen polysulfides in cellular imaging [62,175,176,177,178,179].

Moreover, many studies have been conducted regarding the application of chemosensors based on imidazo[1,5-*a*]pyridines for the quantification of hydrogen sulfide [62] and hydrogen polysulfides [175], sulfur dioxide [177,180], hypochlorous acid [181,182,183], and metal ions such as Cd^2+^ and Zn^2+^ [184], Hg^2+^ [47,185], and Cu^2+^ [170,186].

It is well known that hydrogen ions exert a crucial influence on various aspects of cellular function, including enzymatic and metabolic activity, proliferation, and tumor growth. Therefore, monitoring hydrogen ions through chemosensors holds particular significance. Compounds derived from imidazo[1,5-*a*]pyridines have shown promise as effective ratiometric pH sensors, capitalizing on dual emission for accurate detection [187,188]. Many detection strategies have been tested and employed to assess the exact value of intracellular pH in different cellular compartments and organelles. Probes based on imidazo[1,5-*a*]pyridines have many advantages, such as good selectivity, sensitivity and solubility, and sometimes real-time monitoring. Probes based on imidazo[1,5-*a*]pyridines have been previously reported as fluorescent probes for pH in living cells [59,168].

In addition to the internal charge transfer (ICT) process employed by a single emitter to achieve ratiometric variations, Förster resonance energy transfer (FRET) stands out as one of the most extensively investigated and utilized mechanisms for ratiometric luminescent probes. FRET operates through a non-radiative mechanism where an excited emitter transfers energy to a chromophore acceptor in its ground state. Emitters such as rhodamine, fluorescein, quinine, BODIPY, xanthene, and coumarin have been utilized in the design and synthesis of FRET systems. Typically, the emission spectrum of the donor must substantially overlap with the absorption spectrum of the acceptor, a condition that often poses serious constrains to the development of FRET-based coupled systems. To address this limitation, several research groups have recently developed and synthesized novel imidazopyridinic probes, showcasing their potential for practical FRET-based applications in biological systems [47,177,181,189].

### 2.4. Imidazopyridines as Luminescent Probes in Energy Conversion Technology

For potential applications in optoelectronic devices or down-shifting technologies, luminescent molecules require specific characteristics, notably a broad Stokes shift to prevent reabsorption and a high quantum yield. Generally, absorbed photons should be re-emitted rather than dissipated as heat. A broad Stokes shift minimizes reabsorption, while a high quantum yield ensures that almost every absorbed photon is re-emitted and effectively utilized.

Compact luminescent molecules with a broad Stokes shift are increasingly appealing due to their potential as eco-friendly and cost-effective down-shifting luminescent materials in lighting and photovoltaic applications. Derivatives of imidazopyridines, with their inherent properties, are especially well suited for employment in these technological fields. These compounds have been studied for a large variety of optical applications, including optoelectronic devices [56,57,147], non-linear optical (NLO) studies [14,190,191], and down shifting [192,193].

G. Albrecht et al. recently conducted innovative investigations into the electroluminescence of imidazopyridines [194,195]. They employed layered OLEDs utilizing this fluorogenic core as an emission system [39,40,53,54]. So far, only copper complexes have been successfully applied in light-emitting electrochemical cells. However, several silver, iridium, and zinc complexes have been reported for potential applications in light-emitting electrochemical cells, showing good optical behavior, as well as high quantum efficiency and blue emission [133,135,136,138]. Salassa et al. were the first to report on the electrochemical characterization of both free imidazopyridinic ligands and their corresponding complexes. Their investigations demonstrated that the imidazopyridinic core generally undergoes reversible oxidation at low potentials, a conclusion subsequently corroborated by other studies [155,156,158]. In this context, metal complexes based on imidazopyridine ligands have been studied both as photosensitizers and as redox couples for dye-sensitized solar cells (DSSCs) [196,197,198].

Metal-based complexes offer the opportunity to finely adjust the redox potential of resultant devices through the modification of both metal centers and ligands. Within this framework, the modifiability of imidazopyridine ligands, along with their corresponding electronic levels, presents an intriguing prospect for applications in DSSCs as redox mediators or dyes [199].

### 2.5. Imidazopyridines in Fluorescent and Confocal Microscopy

In recent years, numerous new luminescent probes have emerged for utilization in fluorescence microscopy and confocal microscopy. These investigations encompass a variety of aims, including selective penetration into distinct cellular compartments, visualization of regions with varying pH levels through pH-sensitive fluorescent compounds, and examination of tissues with diverse and specific fluorophores exhibiting distinct optical and chemical characteristics. The research in this domain is concentrated on developing luminescent molecules with intense emission, such as the renowned BODIPY and rhodamine. Conversely, owing to their inherent structural adaptability and remarkable optical attributes, imidazopyridinic derivatives emerge as promising contenders for these applications. The characteristic emission of the imidazopyridinic unit typically falls within the 430–520 nm range, accompanied by a broad Stokes shift extending up to 80 nm [14,59,63,200].

Imidazopyridinic derivatives represent an interesting alternative to achieve intense emission with fluorescence-based microscopy and induced selectivity by the possibility of introducing functional groups or bioconjugation at various points of the heterocycle. Furthermore, imidazopyridinic products exhibit good solubility, intense emission, strong pH dependence, high quantum yield, and remarkable biocompatibility. In recent years, imidazopyridinic emitters have been successfully tested for cell imaging using confocal microscopy. Additionally, imidazopyridinic fluorophores have been tested for multichannel or multicolor bioimaging. To this purpose, cells or tissues are usually treated with different dyes and excited at various wavelengths; consequently, it is necessary to merge various fluorescent images collected under each suitable excitation wavelength for each fluorophore. In this context, imidazopyridinic derivatives have demonstrated the ability to achieve multicolor bioimaging using a single excitation wavelength. This methodology allows for the visualization and detection of reaction kinetics and biomolecular interactions concurrently. As a result, imidazopyridine emitters have been effectively employed in cell imaging utilizing confocal microscopy [59,168,172,174,179].

The comprehensive findings highlight that imidazopyridinic compounds possess sufficient permeability across plant plasma membranes and cell walls, exhibit high adsorption, and distribute effectively within plant internal tissues. Notably, they maintain robust fluorescence in aqueous environments, a crucial attribute for fluorophores designed for in vivo probing. Furthermore, imidazopyridinic compounds have demonstrated effectiveness in distinctly delineating the cytoplasm of mouse fibroblast cells in vitro, suggesting their potential suitability as counterstains to enhance contrast in multicolor fluorescence microscopy [168,187,188,201,202].

## 3. Imidazopyridines: Different Synthetic Approaches

While sharing the generic name of imidazopyridines, each nucleus described above is obtained through different synthetic approaches, ensuring the formation of these heterocycles, which are distinguished by the varying position of nitrogen atoms on the two aromatic rings of imidazole and pyridine. To achieve these structural modifications, it is of primary importance to be aware of the various synthetic approaches available to introduce functional groups, spacers, or additional conjugated systems.

Several reviews have recently been published regarding the synthesis of imidazo[1,5-*a*]pyridines [6,12,46] and imidazo[1,2-*a*]pyridines [43,82,83], the two most studied imidazopyridinic nuclei; relatively few reviews have been published on other imidazopyridine derivatives, but various synthetic approaches have been reported to obtain differently substituted products. Imidazo[1,2-*a*]pyridines are primarily obtained through cyclocondensation reactions, oxidative cyclization, rearrangement reactions, oxidative coupling and multicomponent reactions (see Figure 6) [45].

In the last ten years, significant developments have been made in the synthesis of imidazo[1,2-*a*]pyridines [7,43,45,73,82,203]; however, most of these methods still use a single base reagent, namely 2-aminopyridine, which is employed as a coupling partner in most cases.

Coupling reactions involve various other reagents such as ketones, aldehydes, substituted alkenes, and alkynes (see Figure 6 and Figure 7). Tandem reactions utilize 2-aminopyridine and nitroalkenes as reactants, while multicomponent reactions employ 2-aminopyridine with aldehydes and nitroalkanes or isonitriles or alkynes. Also of interest is the direct reaction between alcohols and 2-aminopyridine, which involves amination followed by cyclization and isomerization [43,45]. Oxidative coupling–cyclization reactions involve the use of copper, silver, or palladium-based catalysts, an oxidizing agent (often atmospheric oxygen), during the reaction of 2-aminopyridine, and other reagents bearing nitrogen-containing functional groups such as nitriles, nitroalkenes, and others [204,205].

Regarding imidazo[1,5-*a*]pyridines, the earliest references involve Vilsmeier-type cyclizations [206]. Subsequently, numerous derivatives of imidazo[1,5-*a*]pyridines (with one or two electron-attracting substituents at positions 1, 2, or 3), were obtained from substituted 2-aminomethylpyridines and phosgene (or ethyl chloroformate) by G. Palazzo and G. Picconi [207].

In 1986, A.P. Krapcho and J.R. Powell employed the same synthetic method as an efficient synthetic approach to collect 1,3-disubstituted imidazo[1,5-*a*]pyridines by treating ketimines derived from commercial 2,2′-dipyridylketone or benzophenone [9]. Interestingly, these ketones have been previously employed to obtain many other ligands with structures different from imidazopyridines [157,208,209,210,211].

More recently, imidazo[1,5-*a*]pyridines are mainly synthesized through condensation reactions (see Figure 8 and Figure 9). Over the years, various catalytic and non-catalytic methods have been developed.

The main condensations involve:Substituted pyridyl-2-ketones with aromatic aldehydes and ammonium acetate (see Figure 9) [103,126,193,212,213,214];Substituted picolylamines (2-(Aminomethyl)pyridin) with carboxylic acids in strongly acidic and dehydrating conditions [9];Other couplings involve cyano-pyridines with aromatic aldehydes, picolylamines with thioesters, and cyclizations of 2-picolylamides [2,46,215,216].

Other synthesis mechanisms involve a cycloaddition approach using benzyl isonitriles as reagents or oxidative cycloadditions using pyridyl-2-ketones and amines (including natural amino acids) or 2-pyridine aldehydes and amines catalyzed by copper oxide and oxygen as oxidant [42,217,218,219].

Several alternative synthesis mechanisms include oxidative cyclization, rearrangement, photochemical reactions, and electrosynthesis. Presently, more than twenty different synthetic strategies have been delineated to achieve this cyclization process. Each unique synthetic approach offers distinct advantages and complementary features for synthesizing imidazo[1,5-*a*]pyridine derivatives.

In recent years, continuous efforts have been dedicated to this fascinating heterocyclization, since the pioneering work of Vilsmeier-type cyclization [206]. However, the substrates employed appear to be limited to specific ketones, nitriles, amides, aldehydes, and substituted N-heteroaryl pyridotriazoles. Additionally, for these transformations, transition metal-based catalysts and additional oxidants were generally required, inevitably leading to the generation of toxic waste and undesired reactions.

Many of the developed synthetic approaches encountered certain limitations, often associated with the use of various catalysts such as Se(IV) [38,220,221], Pd(II) [222,223], Fe(II) and Fe(III) [224,225], Cu(II) ion [226,227,228,229], CuCl_2_ [230,231,232], copper/iodine [233], Cu-MOF structures [234,235], boron trifluoride etherate [236], CBr_4_ [237] or MnO_2_ [53]. In addition, some catalysts require further reagents or co-catalysts such as selenium dioxide [238], dehydrating agents (e.g., molecular sieves [239] or propane phosphoric acid anhydride [213]), Lewis acids (BF_3_), or oxidant agents (e.g., oxygen, iodine [240,241], elemental sulfur [242], or *tert*-butylperoxybenzoate) [243].

Other important examples of innovative synthetic approaches for particular imidazopyridines have been recently reported, focusing on the following:imidazo[1,2-*a*]quinoline [222,244,245];imidazo[1,2-*c*]quinazoline [234];imidazo[1,5-*a*]quinoline [246,247];imidazo[1,5-*a*]pyridine [226,235,248,249,250,251,252];imidazo[1,2-*a*]isoquinoline [221];imidazo[1,5-*a*]isoquinoline [36,42,63,247];imidazo[5,1-*a*]isoquinoline [253,254];imidazo[4,5-*c*]pyridine [28,255,256].

## 4. Summary and Outlook

Over the past forty years, the family of imidazopyridines has significantly expanded its membership, including extended heterocycles such as imidazo[1,2-*a*]quinolines and imidazo[1,5-*a*]quinolones and corresponding isoquinoline. The application of imidazopyridine products ranges from sensing to pharmaceuticals and from materials science to solar energy conversion, optoelectronics, and imaging. Each group of imidazopyridines seems to have taken on a specific role in different application fields; imidazo[1,2-*a*]pyridines have certainly sparked considerable interest in pharmaceuticals, with several commercial products based on this particular heterocycle. Countless studies demonstrate the activity of imidazo[1,2-*a*]pyridines as cardiotonic, antifungal, anti-inflammatory, antitumor, antiviral, antibacterial, antiprotozoal, antipyretic, analgesic, antiapoptotic, and hypnoselective agents.

Conversely, imidazo[1,5-*a*]pyridines owe their interest to their intense and tunable fluorescence emission. The ease with which these heterocycles can be modified and functionalized ensures the ability to modulate absorption and emission wavelength, quantum yield, and lifetime, all crucial parameters for applications in fluorescence microscopy, confocal microscopy, optoelectronics, and solar energy conversion technologies.

Sensor technology can be considered the latest application field of this interesting family; imidazo[1,2-*a*]pyridines and imidazo[1,5-*a*]pyridines have been studied to develop molecular sensors based on on–off fluorescence emission. These fluorophores can act as sensors for pH or metallic ions (such as copper, mercury, or cadmium) or small molecules or ions (sulfides, hypochlorite, cysteine) depending on the functions introduced on these particular aromatic heterocycles.

All imidazopyridines can act as ligands for metallic ions, either as monodentate ligands or, following appropriate functionalizations, as chelating or polydentate ligands. Metallic complexes based on imidazopyridines have been synthesized with all first- and second-row transition metals. Their use ranges from catalysis to electrochemiluminescence; some complexes have been used in OLED technology, confocal microscopy, and solar energy conversion technology. Imidazopyridines certainly represent an excellent basis for the development of fluorescent probes for imaging. Their ease of functionalization, low synthetic costs, along with good biocompatibility, stability, and solubility, make them excellent fluorophores for use in fluorescence microscopy, confocal microscopy, sensing, and quantitative method development.

The future prospects of this versatile compounds’ family certainly involve further exploration of less studied imidazopyridines such as imidazo[4,5-*c*]pyridines, imidazo[4,5-*b*]pyridines, imidazo[5,1-*a*]quinolines, and imidazo[2,1-*a*]isoquinolines, all recently synthesized heterocycles with fewer studies regarding their properties and interactions with other compounds.

Regarding synthesis, methods have recently been developed that exclude the use of expensive or toxic/rare metal-based catalysts. Synthetic strategies have been proposed in water and solvent-free conditions, starting from economical and readily available compounds such as natural amino acids, organic acids, and even aldehydes and ketones of natural origin (such as cuminaldehyde, vanillin, and papaverine). Commercial utilization of these products must involve low-cost, low-impact, highly sustainable synthesis, and recent publications seem to support this possibility.

Imidazopyridines thus describe an extensive family of heterocyclic compounds with very diverse properties, applicable in various technological fields due to their unique biological and photophysical properties. They represent an incredibly interesting group of promising compounds whose study and development could bring significant benefits in many application fields from pharmaceuticals to sensing and from imaging to light energy conversion.

A multidisciplinary approach is necessary to appreciate the versatility of this broad family, which is not found in nature and obtained through various synthesis approaches, from condensation to photocatalytic or microwave-mediated cyclization, and many other methods. Different skills are needed to correctly design the synthesis of imidazopyridine probes, which can be used as sensors, drugs, biological modulators, metallic ion ligands, fluorophores, and many other possible and future application fields. The versatility of these particular molecules is certainly guaranteed by their ease of functionalization, conjugation, and structural modification; however, it is the generic imidazopyridine core that ensures the versatile properties that make this family unique and extremely interesting.

## Figures and Tables

**Figure 1 molecules-29-02668-f001:**
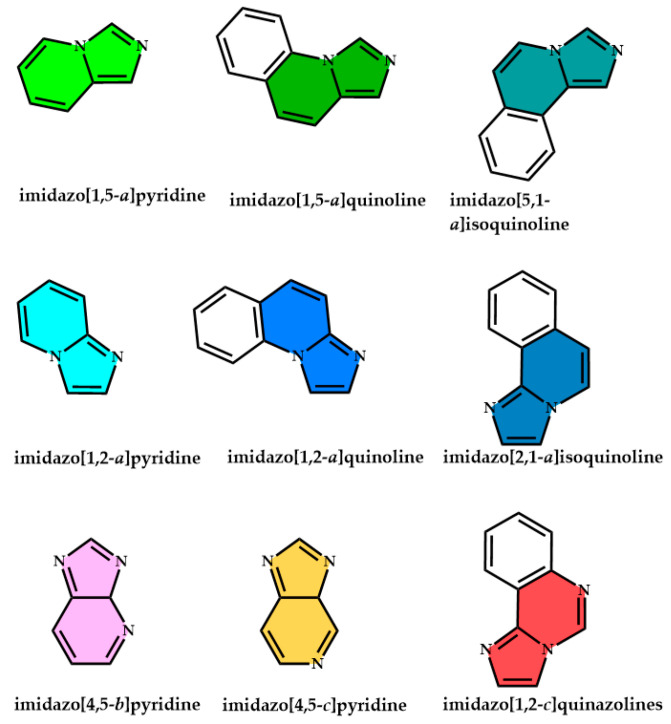
Imidazopyridine family: principal reported heterocyclic skeletons obtained by the union of imidazole and pyridine units.

**Figure 2 molecules-29-02668-f002:**
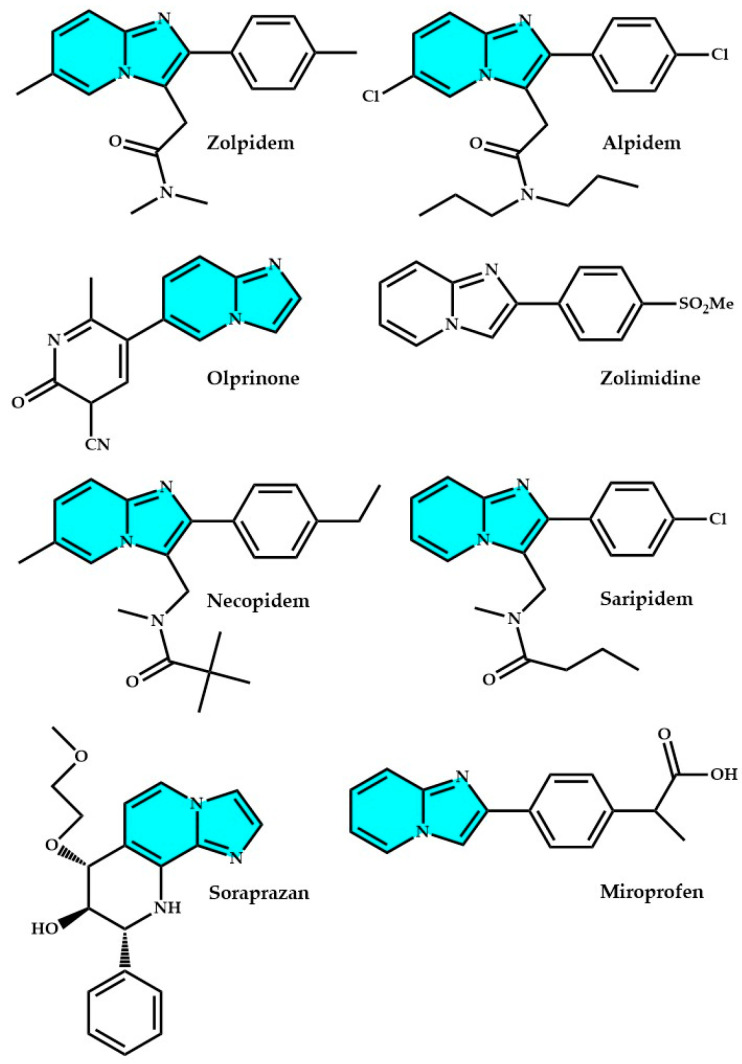
Main imidazo[1,2-*a*]pyridine-based drugs (the imidazo[1,2-*a*]pyridine core is represented here in blue).

**Figure 3 molecules-29-02668-f003:**
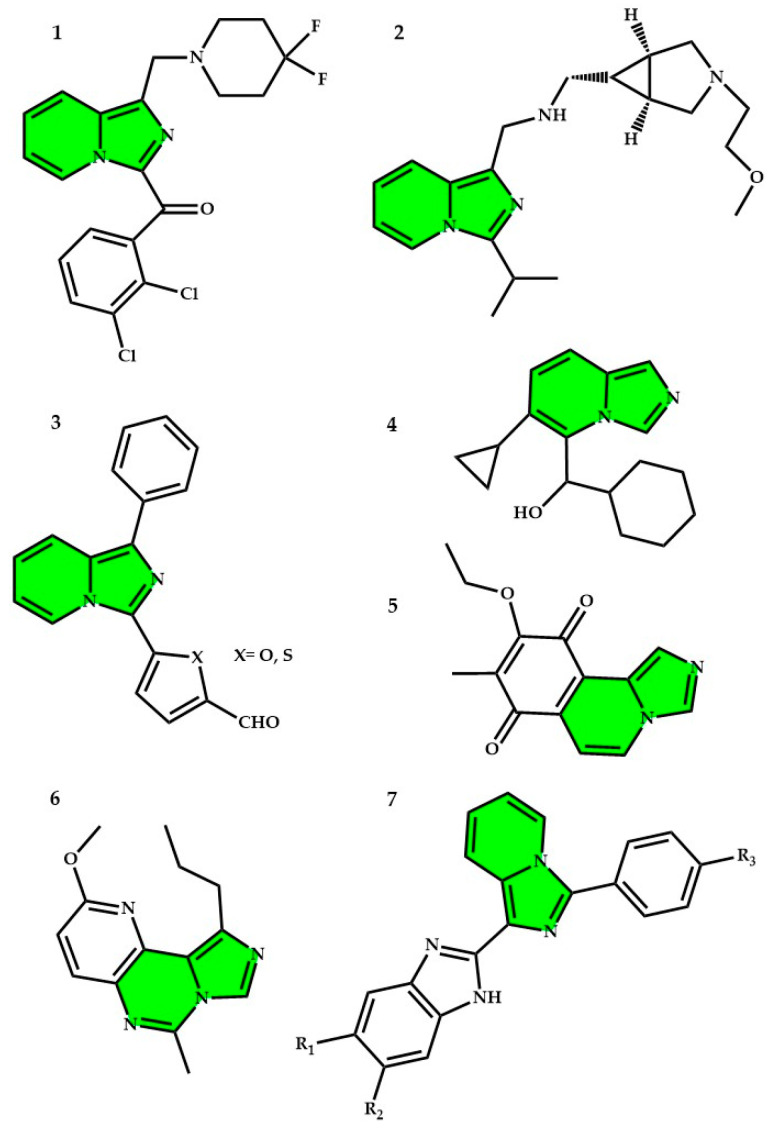
Principal imidazo[1,5-*a*]pyridine biologically active derivatives (the imidazo[1,5-a]pyridine core is represented here in green, the names of the represented products are listed in the text according to the given numbers).

**Figure 4 molecules-29-02668-f004:**
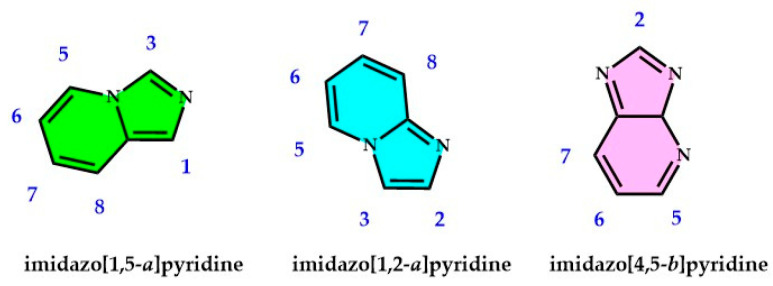
Principal imidazopyridine nuclei and the corresponding positions for possible derivatization and introduction of substituent groups (different imidazopyridine cores filled with the colors used in Figure 1).

**Figure 5 molecules-29-02668-f005:**
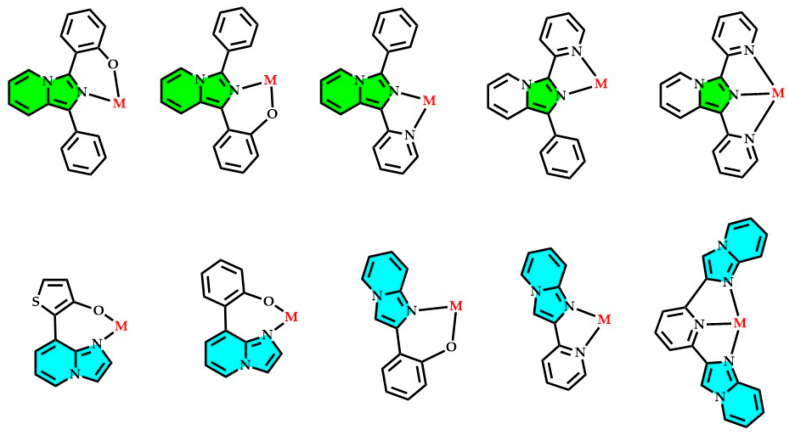
Main structures of imidazo[1,5-*a*]pyridinic (green) and imidazo[1,2-*a*]pyridinic (blue) derivatives employed as N-O, N-N, or multi-dentate ligands.

**Figure 6 molecules-29-02668-f006:**
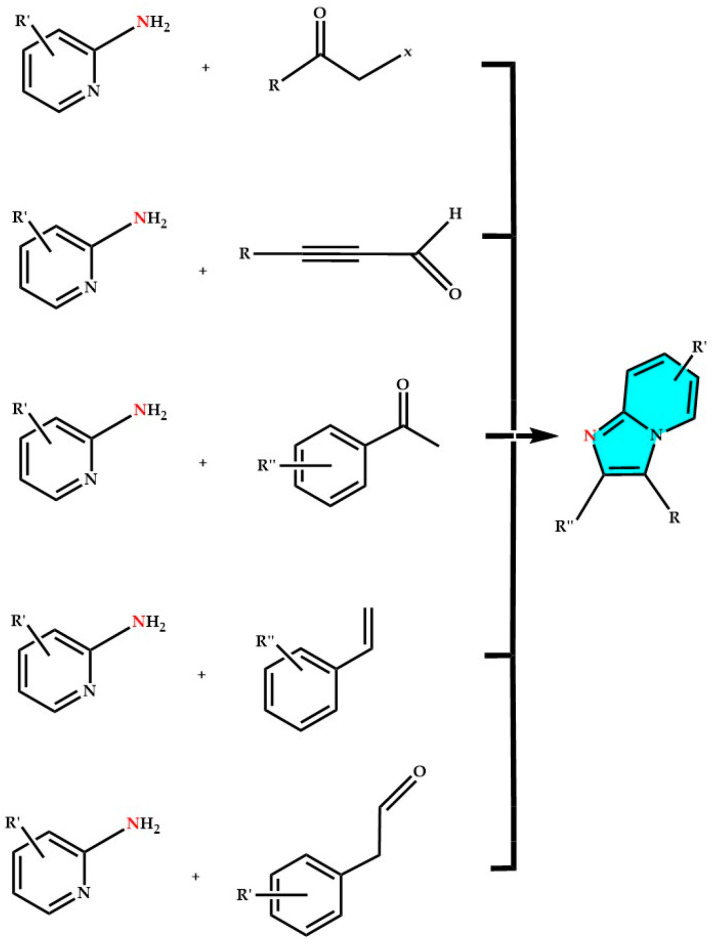
Early reported synthetic approaches to obtain substituted imidazo[1,2-*a*]pyridine core (blue), inspired by [45,82].

**Figure 7 molecules-29-02668-f007:**
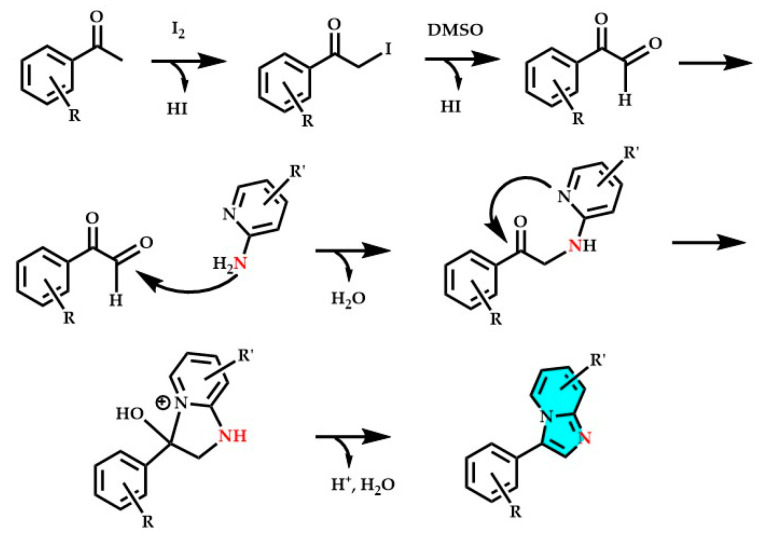
Complete reaction mechanism of one of the most used synthetic approaches involving substituted ketones and 2-aminopyridine to obtain imidazo[1,2-*a*]pyridine derivatives (blue), mechanism reported in [43,45].

**Figure 8 molecules-29-02668-f008:**
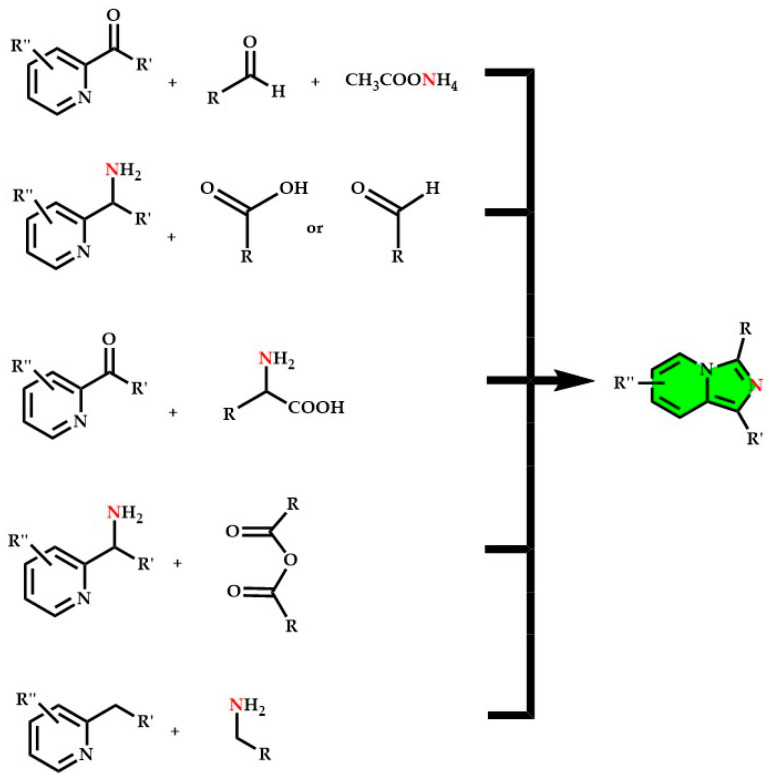
Early reported synthetic approaches to obtain substituted imidazo[1,5-*a*]pyridine core (green), inspired by [12].

**Figure 9 molecules-29-02668-f009:**
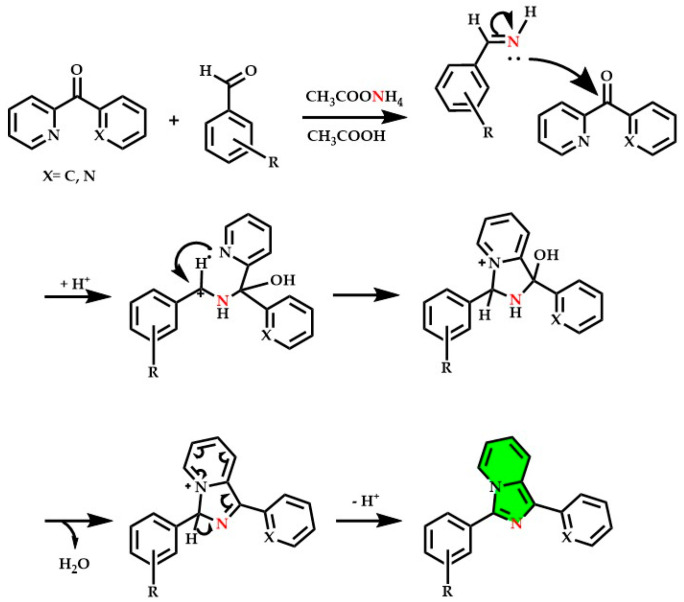
Complete reaction mechanism of one of the most used synthetic approaches to obtain imidazo[1,5-*a*]pyridine derivatives involving aldehydes and pyridyl-2-ketones, mechanism reported in [12,126].

## Data Availability

The data presented in this study are available within the article.

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
