# Peer review of "Imidazopyridine Family: Versatile and Promising Heterocyclic Skeletons for Different Applications"

_molecules, 2024, doi:10.3390/molecules29112668_

Round 1

Reviewer 1 Report

Comments and Suggestions for Authors

This manuscript summarizes the development in imidazopyridine family. Authors have presented a concise summary of various applications associated with these heterocycles. The topic is interesting and could be considered after following corrections. My comments are

1. Mention the time frame in abstract for literature covered.

2. Abstract could be further populated with key applications.

3. Remove the filled heterocycles throughout the manuscript. Keep it simple and highlight with different colors rather than filling the space.

4. Chemical structures are not uniform and are blurred. 

5. Synthetic strategies are limited. Include further literature on it. Citing few references in one line is not sufficient.

5. Why authors did not include the medicinal chemistry properties of these heterocycles? These are valuable medicinal scaffolds. Take a look, otherwise modify the title and accommodate the presented applications accordingly.

Comments on the Quality of English Language

Minor editing required otherwise good.

Author Response

REPORT 1

Comments and Suggestions for Authors

This manuscript summarizes the development in imidazopyridine family. Authors have presented a concise summary of various applications associated with these heterocycles. The topic is interesting and could be considered after following corrections. My comments are:

  1. Mention the time frame in abstract for literature covered.
  • We thank referee 1 for this important clarification. The follow sentence was added to the maintext: "This review intends to provide a state-of-the-art overview of this framework from 1955 to the present day, unveiling different aspects of various applications. This extensive literature survey may guide chemists and researchers in the quest for novel imidazopyridine compounds with enhanced properties and efficiency in different uses.”

  1. Abstract could be further populated with key applications.
  • We thank referee 1 for this observation. The manuscript has been modified by adding the following paragraph: “The various cores belonging to the imidazopyridine family exhibit unique properties, such as serving as emitters in imaging, ligands for transition metals, showing reversible electrochemical properties, and demonstrating biological activity.”
  1. Remove the filled heterocycles throughout the manuscript. Keep it simple and highlight with different colors rather than filling the space.
  • We thank referee 1 for the suggestion. All figures have been redone, standardizing style, dimensions, and text font. But we request the editor to maintain this particular graphical choice to better highlight the different imidazopyridine cores. Indeed, the same style has already been used in previous works on the same subject (i.e. Biomol. Chem., 2022, 20, 4215 or Org. Biomol. Chem., 2023, 21, 9552-9561 or Molecules 2022, 27, 3461 or Molecules 2024, 29, 653 and Talanta 2020, 217, 121087), where different types of imidazopyridines are distinguished using different colors.

  1. Chemical structures are not uniform and are blurred.
  • We thank Referee 1 for this observation. We have revised all the figures, ensuring uniform dimensions, graphical style, and using the font of the main text.
  1. Synthetic strategies are limited. Include further literature on it. Citing few references in one line is not sufficient.
  • We thank Referee 1 for this comment. The aim of this work is to present the applications of various cores within the extensive family of imidazopyridines. Rather than focusing on different synthetic approaches, this study highlights the application fields and uses of these compounds. Numerous articles and specific reviews on synthesis, which report various synthetic approaches and reaction mechanisms for obtaining imidazo derivatives, are cited in paragraph "3. Imidazopyridines: Different Synthetic Approaches." It is important to highlight that the various imidazopyridine nuclei are synthesized in completely different ways. For this reason, synthetic reviews always focus on a single type of imidazopyridine. In contrast, the aim of this work is to provide a comprehensive overview of the applications of imidazopyridines, which are often very diverse from each other.

  1. Why authors did not include the medicinal chemistry properties of these heterocycles? These are valuable medicinal scaffolds. Take a look, otherwise modify the title and accommodate the presented applications accordingly.

We thank Referee 1 for the important observations. In the presented work, we have reported the main uses of imidazopyridines in the pharmaceutical and medical fields. In addition, the text has been revised, and two new paragraphs have been added to section “2.1 Imidazopyridines in medicine” concerning medical applications. Moreover, 15 new references have been added, mostly related to recent publications (2023-2024), focusing on the use of imidazopyridines for cancer treatment. It is important to highlight that within the extensive family of imidazopyridines, only imidazo[1,2-a]pyridine products currently have real applications in the pharmaceutical and medical fields. Therefore, this manuscript has aimed to dedicate substantial space to other imidazopyridine cores in different application fields.

Reviewer 2 Report

Comments and Suggestions for Authors

1. What is the main question addressed by the research?

 The review entitled “Imidazopyridine family: versatile and promising heterocyclic skeletons for different applications” is dedicated to a family of imidazopyridine compounds, which possess unique structural features and diverse optical and biological properties. The imidazopyridine core is known to be one of the privileged pharmacophore scaffolds that can be found in numerous biologically active compounds. Much effort has been invested into the development of efficient methods for the synthesis of imidazopyridines. Currently more than 120,000 individual representatives of this family are known. Therefore, the relevance and significance of the paper summarizing main modern trends in the chemistry of imidazopyridines is beyond dispute. In opinion to this reviewer, the work will be of interest to the experts in organic chemistry, pharmaceutics and biology, as well as to specialists who are dealing with new trends in the chemistry of N-heterocyclic compounds.

2. What parts do you consider original or relevant for the field? What specific gap in the field does the paper address?

 The originality of the work is contingent upon a large body of references, covering mainly the last 25 years, but earlier works that have made significant contributions to the field are also surveyed.

3. What does it add to the subject area compared with other published material?

In the review, various model syntheses of compounds containing imidazopyridine fragments are summarized. Both non-catalytic and catalytic approaches are discussed. It should be noted that organometallic compounds, MOF structures, Lewis acids and complex catalytic systems are considered as catalysts. In addition, the application fields of different imidazopyridines are surveyed.

4. What specific improvements should the authors consider regarding the methodology? What further controls should be considered?

Unfortunately, the advantages and shortcomings of each specific synthetic protocol are discussed insufficiently in the text. Also, compounds with imidazopyridine moieties are known to be often employed in the synthesis of catalytically active structures. However, this fact is not highlighted in detail in the paper.

5. What specific improvements should the authors consider regarding the analysis and discussion? How can the authors revise to improve this work?

 The font size in all figures should be unified and should correspond to the font size in the main text. All figures should be made in one style. Under each scheme or figure, reference(s) to original publication should be given, even if these schemes or figures are the authors’ own work.

Misprints should be corrected. For example, the red line is missing (lines 456, 553, 564, 574, etc.), in lines 110 and 118 the presence of red lines is implied, but the paragraph indent is too small, punctuation marks are absent (e.g. line 480) , dots are not placed at the end of section names, some abbreviations are not defined, e.g., GABA.

6. Please describe how the conclusions are or are not consistent with the evidence and arguments presented. If not, what contents/analysis needs to be added to improve it?

The conclusions are quite consistent with the evidence and arguments of the work. It might be useful if the authors discuss in more detail the medical application imidazopyridines, since currently such compounds are most popular in medicine.

7. Are the references appropriate?

The references are appropriate and sufficiently reflect the model examples discussed in the paper.

Author Response

REPORT 2

Comments and Suggestions for Authors

  1. What is the main question addressed by the research?

 The review entitled “Imidazopyridine family: versatile and promising heterocyclic skeletons for different applications” is dedicated to a family of imidazopyridine compounds, which possess unique structural features and diverse optical and biological properties. The imidazopyridine core is known to be one of the privileged pharmacophore scaffolds that can be found in numerous biologically active compounds. Much effort has been invested into the development of efficient methods for the synthesis of imidazopyridines. Currently more than 120,000 individual representatives of this family are known. Therefore, the relevance and significance of the paper summarizing main modern trends in the chemistry of imidazopyridines is beyond dispute. In opinion to this reviewer, the work will be of interest to the experts in organic chemistry, pharmaceutics and biology, as well as to specialists who are dealing with new trends in the chemistry of N-heterocyclic compounds.

  • We thank referee 2 for the positive evaluation of the manuscript.
  1. What parts do you consider original or relevant for the field? What specific gap in the field does the paper address?

 The originality of the work is contingent upon a large body of references, covering mainly the last 25 years, but earlier works that have made significant contributions to the field are also surveyed.

  • We thank referee 2 for the evaluation of themaniscript. The maintext has been modified as follows: "This review intends to provide a state-of-the-art overview of this framework from 1955 to the present day, unveiling different aspects of various applications. This extensive literature survey may guide chemists and researchers in the quest for novel imidazopyridine compounds with enhanced properties and efficiency in different uses.”

  1. What does it add to the subject area compared with other published material?

In the review, various model syntheses of compounds containing imidazopyridine fragments are summarized. Both non-catalytic and catalytic approaches are discussed. It should be noted that organometallic compounds, MOF structures, Lewis acids and complex catalytic systems are considered as catalysts. In addition, the application fields of different imidazopyridines are surveyed.

  • We thank Referee 2 for this comment. The aim of this work is to present the applications of various cores within the extensive family of imidazopyridines. Rather than focusing on different synthetic approaches, this study highlights the application fields and uses of these compounds. Numerous articles and specific reviews on synthesis (which report various synthetic approaches and reaction mechanisms for obtaining imidazo derivatives) are cited in paragraph "3. Imidazopyridines: Different Synthetic Approaches.

  1. What specific improvements should the authors consider regarding the methodology? What further controls should be considered?

Unfortunately, the advantages and shortcomings of each specific synthetic protocol are discussed insufficiently in the text. Also, compounds with imidazopyridine moieties are known to be often employed in the synthesis of catalytically active structures. However, this fact is not highlighted in detail in the paper.

  • We thank Referee 2 for this comment. The aim of this work is to present the applications of various cores within the extensive family of imidazopyridines. Rather than focusing on different synthetic approaches, this study highlights the application fields and uses of these compounds. Numerous articles and specific reviews on synthesis, which report various synthetic approaches and reaction mechanisms for obtaining imidazo derivatives, are cited in paragraph "3. Imidazopyridines: Different Synthetic Approaches." It is important to highlight that the various imidazopyridine nuclei are synthesized in completely different ways. For this latter reason, the cited paragraph aims to provide references to guide readers interested in the synthesis of such compounds to recent literature, while the purpose of the manuscript is to inform about their use in various technological fields. While in the literature there are many references dedicated to the preparation of such compounds, there are no articles on the applications of the entire family of imidazopyridines. Therefore, the manuscript focuses on this important aspect.

  1. What specific improvements should the authors consider regarding the analysis and discussion? How can the authors revise to improve this work?

 The font size in all figures should be unified and should correspond to the font size in the main text. All figures should be made in one style. Under each scheme or figure, reference(s) to original publication should be given, even if these schemes or figures are the authors’ own work.

  • We thank Referee 2 for this observation. We have revised all the figures, ensuring uniform dimensions, graphical style, and using the font of the main text. All the figures are specially realized for this paper by the authors. However, some of them (figures 6-9) are inspired by previously published synthetic approaches or reaction mechanisms and the relative references were added in the caption as suggested by the referee.

Misprints should be corrected. For example, the red line is missing (lines 456, 553, 564, 574, etc.), in lines 110 and 118 the presence of red lines is implied, but the paragraph indent is too small, punctuation marks are absent (e.g. line 480) , dots are not placed at the end of section names, some abbreviations are not defined, e.g., GABA.

  • We thank Referee 2 for the provided comments. The text has been revised, and in accordance with the template provided by the publisher, the dots at the end of the section titles has been removed. We request the publisher to inform us of any necessary changes during the final editing stage. Moreover, all abbreviations have been spelled out in the main text (e.g., OLED, TDDFT, GABA, etc.).

  1. Please describe how the conclusions are or are not consistent with the evidence and arguments presented. If not, what contents/analysis needs to be added to improve it?

The conclusions are quite consistent with the evidence and arguments of the work. It might be useful if the authors discuss in more detail the medical application imidazopyridines, since currently such compounds are most popular in medicine.

  • We thank referee 2 for his valuable suggestion. We have added several paragraphs concerning the medical applications of imidazopyridine nuclei. Specifically, we introduced a paragraph on imidazo[1,2-b]pyridazines an important class of bioactive molecules with potential therapeutic applications in medicine (including anticancer agents, diagnostic tools for neuropathic diseases, thymic enhancers, antiparasitic, antibacterial, and antiviral agents, anti-inflammatory agents, and treatments for circadian rhythm sleep disorders).

Additionally, we have added two paragraphs on the potential of imidazo[1,2-a]pyridines, which have been efficiently employed to detect amyloid pathology in the brains of patients with Alzheimer’s disease, as well as recent studies reporting their antiproliferative activity against lung, breast, and cervical cancer cells. In the revised version of the manuscript, 15 new references related to the medical applications of imidazopyridines from the period 2023-2024 have been added.

  1. Are the references appropriate?

The references are appropriate and sufficiently reflect the model examples discussed in the paper.

  • We thank the referee for their feedback. Additionally, we have added 15 new references following the requested revisions.

Round 2

Reviewer 1 Report

Comments and Suggestions for Authors

Authors have improved the manuscript. It could be considered now.

Comments on the Quality of English Language

Minor editing required